# Identification of *Chelonus* sp. from Zambia and Its Performance on Different Aged Eggs of *Spodoptera frugiperda*

**DOI:** 10.3390/insects14010061

**Published:** 2023-01-09

**Authors:** Zhen Shen, Zhuo-Yi Zang, Peng Dai, Wei Xu, Phillip O. Y. Nkunika, Lian-Sheng Zang

**Affiliations:** 1College of Plant Protection, Jilin Agricultural University, Changchun 130118, China; 2Department of Biological Sciences, School of Natural Sciences, University of Zambia, Lusaka 10101, Zambia; 3Key Laboratory of Green Pesticide and Agricultural Bioengineering, Ministry of Education, Guizhou University, Guiyang 550025, China

**Keywords:** *Spodoptera frugiperda*, egg parasitoid, *Chelonus*, biological control, Africa

## Abstract

**Simple Summary:**

An egg-larva endoparasitoid (*Chelonus bifoveolatus*) of fall armyworm (FAW) in Zambia was identified based on morphological and molecular characteristics. To assess the parasitism capabilities of *C. bifoveolatus* against FAW, we compared partial biological parameters parasitizing 0- to 2-day-old FAW eggs. *Chelonus bifoveolatus* successfully parasitized and developed on eggs in all tested samples. In addition, it had a higher parasitism rate, pupation rate, and emergence rate on 1-day-old FAW eggs, but shorter development time on 2-day-old FAW eggs. Eggs of all ages tested revealed that females to males sex ratio was nearly 1:1.

**Abstract:**

The fall armyworm (FAW), *Spodoptera frugiperda* (J. E. Smith) (Lepidoptera: Noctuidae), is a migratory pest endemic, to tropical and subtropical regions of America. Biological control can effectively and sustainably control pests over a long period of time while reducing the frequency of pesticide use and ensuring the safety of agricultural produce. In our study, the egg-larval *Chelonus* species (*Chelonus bifoveolatus*) from parasitized eggs of *Spodoptera frugiperda* in Zambia were described and identified based on morphological and genetic characteristics. To evaluate the efficiency of *C. bifoveolatus*, their parasitism suitability on 0- to 2-day FAW eggs under laboratory conditions was compared. The results showed that *C. bifoveolatus* could accept all FAW eggs at 0-, 1- and 2-day-old age and complete development successfully. Significant differences were found among 0-, 1-, and 2-day-old host eggs with respect to egg-larva developmental duration of *C. bifoveolatus*, and the egg-larva developmental duration on 2-day-old eggs was significantly lower than those on 0- and 1-day-old eggs. No significant differences were observed in the parasitism, pupation, emergence, and female rates for *C. bifoveolatus* on various age eggs of FAW. Generally, the parasitism rate, pupal rate, and emergence rate at various ages of FAW eggs were higher than 90%, 75%, and 82%, respectively, and the longevity of female parasitoids was longer than male parasitoids, and the sex ratio of females to males was nearly 1:1. Our results indicate that *C. bifoveolatus* performed well on various ages of FAW eggs and is a potential biological control agent against FAW in Africa.

## 1. Introduction

The fall armyworm (FAW), *Spodoptera frugiperda* (J. E. Smith) (Lepidoptera: Noctuidae), a migratory pest endemic to tropical and subtropical regions of America was first recorded on grains and grasses in Georgia in 1797 [1,2,3]. The following attributes are associated with the first report in West Africa, wide adaptability, strong migration ability, high fecundity, lack of diapause, quick resistance to insecticides, and spread to more than 60 countries and regions, among which Africa, Asia, and Oceania [3,4,5,6]. The FAW larvae feed on leaves, stems, and reproductive parts of a wide range of host plants, including cultivated crops, such as maize, wheat, rice, peanut, soybean, cotton, tobacco, and vegetables, among others [7]. Hence, the major socio-economic consequence for human and global food security is under a serious threat [8].

In most African countries, favorable climatic conditions, available host plants, off-season, and irrigated crops allow the pest to complete several generations per year [9,10]. Maize is the most important economic and staple food crop in Africa, where it is grown predominantly by small-holder farmers [11]. The occurrence of fall armyworm in over 44 countries in Africa threats the food security of millions of people [12,13]. In the absence of proper control methods, FAW has the potential to cause maize yield losses of 8.3 m tons to 20.6 m tons per year in just 12 of Africa’s maize-producing countries and economic loss of $2.53 billion to $6.31 billion [13,14]. In Zambia, FAW has the potential to cause maize yield losses of 0.7 m tons to 14.6 m tons per year, which was equivalent to an economic loss of $125.2 million to $250.4 million [14].

Several options are available to mitigate the impact of FAW, such as synthetic insecticides, biopesticides, botanicals, genetically modified crops, mechanical control practices, and cultural control [14,15]. However, the immediate response for the management methods to control FAW has focused primarily on chemical insecticides in Africa, mainly due to the unavailability of alternative and governmental emergency programmers subsidizing synthetic insecticides [14,16,17,18]. As a result, there is an abundant influx of pesticides into fields that previously did not apply pesticides to maize growing fields [19,20]. Frequent and improper use of pesticides can favor the development of insecticide resistance, reduce the population of beneficial natural enemies, and cause environmental problems, and further, can pose a threat to human health [18,21,22,23]. Because of the excessive use of insecticides and small-holder farmers using scientifically unproven methods, such as the application of ash, sand, botanical extracts, and other locally available materials, this damage has created challenges all over Africa [24]. Hence, there is an urgent need for readily available, safe, effective, and sustainable alternative agents [13].

Biological control can effectively and sustainably control pests over a long period of time while reducing the frequency of pesticide use and ensuring the safety of agricultural products [25,26]. Therefore, biological control approaches exploit the use of parasitoids, predators, entomopathogenic viruses, bacteria, and fungi as viable alternatives for the management of this pest [27,28,29,30,31,32,33,34]. Among the approximately 150 species of parasitoids and parasites that have been recorded worldwide, the egg parasitoids *Trichogramma dendrolimi* (Matsumura) (Hymenoptera: Trichogrammatidae) and *Telenomus remus* (Nixon) (Hymenoptera: Platygastridae), the egg—larval parasitoids *Chelonus insularis* (Cresson) and *Chelonus antillarum* (Marshall) (Hymenoptera: Braconidae), the larval parasitoids *Coccygidium luteum* (Brullé) and *Cotesia marginiventris* (Cresson) (Hymenoptera: Braconidae), the pupal parasitoids *Diapetimorpha introita* (Cresson) (Hymenoptera: Ichneumonidae) and *Brachymeria ovata* (Say) (Hymenoptera: Chalcididae), can be used against FAW [35,36,37,38].

The genus *Chelonus* (Hymenoptera: Cheloninae), an egg-larva endoparasitoid, is one of the largest genera of the subfamily Cheloninae (Hymenoptera, Braconidae) with 973 described species worldwide [39] and is the most common and widely distributed parasitoid of FAW in the Americas and Africa [40]. In Colombia, *C. insularis* was reported parasitizing FAW, but is very susceptible to insecticides (e.g., chlorpyriphos, methomyl, and cypermethrin) except for *Bt* toxin [41]. In the Caribbean Islands, *C*. *antillarum* Marshall parasitizes eggs of FAW [42]. In South and West Africa, *C. insularis* caused about 91% of natural parasitism in maize field samples. In Tanzania, the abundance of *C. bifoveolatus* has a significant interaction between the cropping system and pesticide application [37]. However, *Chelonus* spp. reared in the laboratory using factitious host, have been used widely as biological control agents to control *Batrachedra arenosella* (Walker) (Lepidoptera: Batrachedridae), *Earias vittella* (Fabricius) (Lepidoptera: Noctuidae), *Helicoverpa armigera* (Hübner) (Lepidoptera: Noctuidae), *Pectinophora gossypiella* (Saunders) (Lepidoptera: Gelechiidae), *Prays oleae* (Bernard) (Lepidoptera: Praydidae) and *Spodoptera littoralis* (Boisduval) (Lepidoptera: Noctuidae) [43,44,45,46,47]. The aim of our study was to identify the *Chelonus* species, collected from parasitized FAW eggs in Zambia in 2019, using a combination of characteristics of external morphology and molecular techniques. In addition, the performance of *Chelonus* species on FAW eggs of different ages to determine the biological control potential was explored to provide the theoretical basis for their application in the future.

## 2. Materials and Methods

### 2.1. Sample Collection and Insect Rearing

The original FAW eggs (parasitized and unparasitized) were collected from maize fields at China-aid Zambia Agricultural Technology Demonstration Centre (15°21′30″ S, 28°27′27″ E), Lusaka, Zambia, in September of 2019. Each egg mass was kept in a petri dish (diameter = 10.5 cm, height = 2.5 cm) containing fresh maize leaves to ensure that the enclosed neonates had available food and was kept at laboratory conditions with 26 ± 1 °C, R.H. 70 ± 5% and photoperiod 14:10 (L:D) h. The larvae were reared on maize leaves until they became FAW pupae or parasitoid cocoons. After emergence, the FAW adults and parasitoids were introduced into a cage (length, width, height = 35.0 cm) and provided with a cotton-wool ball soaked with 20% honey as diet, respectively. Filter plastic paper was provided for FAW adults as an oviposition substrate and periodically replaced, as required. FAW egg masses attached to plastic paper were introduced into the rearing cage of parasitoid adults for 4 h to allow parasitism. Then, each egg mass attached to plastic paper was placed in a petri dish (diameter = 10.5 cm, height = 2.5 cm) containing a piece of artificial diet that was made from Greene et al.’s method [48]. The larvae were reared individually when they reached the third instar in 12-well plastic plate until they emerged as adults [49].

### 2.2. Morphological Identification

Ten newly emerged female and 10 newly emerged male parasitoids were introduced in a 1.5 mL tube, then put at −40 °C refrigerator for 10 min. The parasitoids were later soaked in 5% potassium hydroxide in a plastic dish (diameter = 2.0 cm) for 24 h and subsequently washed 3–4 times with distilled water. Finally, tissues were separated under the microscope and soaked 5 min in graded alcohols at concentrations of 50, 70, 80, 85, 90, 95, 100%, respectively. Based on morphological characters, the specimen was initially identified using the published taxonomic literature on Braconidae [50,51,52,53,54]. Measurement methods were used according to Van Achterberg [55,56], Harris [57], and Zack [58]. Specimens were examined with a Nikon SMZ 1500 binocular microscope, and photographs were made with a Keyence (VHX-2000) digital microscope. Photographs were slightly processed (mainly cropping and modification of background) in Adobe Photoshop CC 2018.

### 2.3. Molecular Identification

#### 2.3.1. DNA Extraction

Genomic DNA of *Chelonus* samples were extracted following Kumar et al.’s method [59]. Two female adults were put in a 2 mL tube, and three magnetic beads were placed. Tissue that was previously frozen for 5–10 min at −80 °C was crushed 120 s with a grinding apparatus in 200 µL GA. 20 µL proteinase k was added, and the mixture was incubated for three hours at 56 °C, and thereafter, 200 µL GB was added for incubation for 10 min at 70 °C. After incubation, 200 µL of 100% ethanol was added and centrifuged at 12,000 rpm for 30 s. The flocculent precipitate was transferred into the adsorption column CB3 and centrifuged at 12,000 rpm for 30 s. 500 µL of GD was added, followed by centrifugation at 12,000 rpm for 30 s. 600 µL of PW was added, and centrifuged at 12,000 rpm for 30 s and repeated once. The adsorption column CB3 was transferred into another 1.5 mL tube and centrifuged at 12,000 rpm for two minutes, and the pellets were air-dried for several minutes. Overall, 70 µL distilled water was added and stored at −20 °C for further use.

#### 2.3.2. DNA Amplification

DNA barcodes were amplified by the standard barcoding primer pair LCO1490 (5′-GGT CAA CAA ATC ATA AAG ATA TTG G-3′) and HCO2198 (5′-TAA ACT TCA GGG TGA CCA AAA AAT CA-3′) [60]. Polymerase chain reaction (PCR) was conducted in the total volume of 50 µL consisting of 25 µL 2× Taq Premix Mix, 1 µL forward primer and reverse primer at 10 µM of concentration, 2 µL genomic DNA (10–30 ng/µL) and 21 µL distilled water. One cycle of initiation was performed at 94 °C for 5 min, followed by 35 cycles of 94 °C for 30 s, primer annealing at 52 °C for 30 s, primer extension at 72 °C for 30 s. Afterwards, the reactions were kept at 72 °C for 10 min as the final step. The PCR products were separated by electrophoresis in 1× TAE buffer using a 1% agarose gel stained with ethidium bromide (EB) at 150 volts for 30 min. A 2000 bp DNA ladder (Sangon Biotech (Shanghai) Co., Ltd., Shanghai, China) was used as a size marker. The result was visualized under JS-3000 UV transilluminator (Peiqing tech (Shanghai) Co., Shanghai, China). Then PCR products were sent to Sangon (Shanghai) for bidirectional sequencing.

### 2.4. Assessment of Biological Characteristics

FAW egg masses (containing 200–300 eggs) laid on plastic paper were randomly selected from cages of 100–200 adults, and egg masses with different ages (within 1 h as 0-day age, 1- and 2-day age hosts indicated the egg masses that had developed for one and two days, respectively) were prepared under the microscope and individually placed in glass tubes (diameter = 10.5 cm, height = 2.5 cm). Afterwards, one mated female was randomly selected from a cage that had 200 parasitoids (less than 24 h of age, fed with 20% honey water and no oviposition experience), introduced in glass tubes containing egg masses and a cotton ball soaked in 20% honey water, and allowed to oviposit eggs for 24 h. The egg mass was removed and placed in a petri dish (diameter = 6.0 cm, height = 1.5 cm). A few drops of (0.4–0.5 mL) water was added to the absorbent cotton to avoid desiccation. When larvae hatched, a piece of artificial diet was made from Greene et al.’s method [48], and this diet was placed at the bottom of the Petri dishes. Petri dishes were closed at the top with paper. In order to avoid cannibalism among the old larvae, they were individually transferred into 12-well plastic plates with fresh artificial diets. Wells with larvae were tested and checked daily for the developmental stage until emergence or death. The dead larvae were dissected under a microscope to confirm the presence of parasitic wasp larvae. When the parasitized larvae started curling their bodies, an indication that the larvae of parasitoids were about to reach the pre-pupal period, the frass was cleaned, and the larvae were covered with thin cotton until successful pupation. Newly emerged females and males were recorded, and individuals were placed in glass tubes (diameter = 10.5 cm, height = 2.5 cm) covered with gauze element (100 mesh) to allow air exchange. Cotton balls with 20% honey water were placed inside the glass tubes until the parasitoids died. The experiment was conducted under laboratory conditions at 26 ± 1 °C, relative humidity of 70 ± 5%, and photoperiod of 14:10 (L:D) h. There were 5 replicates for various egg ages.
Parasitism rate = (number of parasitoid larvae/total number of larvae) × 100%
Pupation rate = (number of cocoons/total number of parasitoid larvae) × 100%
Emergence rate = (number of parasitoids/total number of cocoons) × 100%

### 2.5. Statistical Analysis

#### 2.5.1. Phylogenetic Analysis

The sequences were retrieved from Basic Local Alignment Search Tool (BLAST) base (http://www.ncbi.nlm.nih.gov/genbank/) (accessed on 20 October 2022) and Barcode of Life and Data-system (Bold) to reveal the possible identity. The phylogenic tree was constructed using the Neighbor-Join and BioNJ algorithms in MEGA 6.0 software to a matrix of pairwise distances estimated using the Maximum Composite Likelihood (MCL) approach and then selecting the topology with superior log likelihood value. Branch support was evaluated by bootstrapping with 1000 repetitions.

#### 2.5.2. Bioassay Analysis

A one-way analysis of variance (ANOVA) was conducted to determine the effect of treatment on the number of parasitized larvae, developmental time, longevity, parasitism rate, percentage of pupation, emergence rate, and percent female progeny. Tukey’s honestly significant difference (HSD) test was used to compare means at *p* < 0.05. All data were subject to a normality test (Shapiro–Wilk test) prior to ANOVA. All percentage data were arcsine square-root-transformed prior to the Shapiro–Wilk test. The analysis was performed on the transformed data, and untransformed means ± SE were presented. Longevity between males and females was analyzed using the independent-samples *t*-test for statistical analysis, with a confidence level of 95%. All statistical analyses were performed using SPSS version 20 software package (SPSS Inc., Chicago, IL, USA).

## 3. Results

### 3.1. Morphological Characteristics

**Description**. Holotype, female adult, length of body 5.0~7.0 mm, forewing 4.0~5.0 mm (Figure 1A, Figure 2A).

**Head.** Head transverse, length of head 2.3 × its width. Antennae 24~26, whip, average length 3rd, 4th, and terminal segments 3.0, 2.5, and 2.1 × their width, respectively. Oval eyes, length of eyes 2.3 × temple in dorsal view, OOL:OD:POL = 17:4:12 (Figure 2G). Vertex Face striate-rugose, covered with white hairs, width of the face is 3.2 times its height (Figure 2F). Clypeus evenly convex and punctate. Temple finely and areolate-rugose, frons concave and longitudinal striate-rugose around antennal sockets. Malar space is 1.4 × base width of the mandible (Figure 2I).

**Mesosoma:** Developed mesosoma with strongly reticulate rugosity, length 1.3 × its height (Figure 2B), pronotum slightly saddle-shaped with irregular rugose, mesonotum strongly convex, mesoscutum evenly convex, reticulate-rugose, mesopleurum coarsely reticulate-rugose, scutellar narrow and rugose, scutellar suture deep and with several carinae (Figure 2C), propodeum coarsely reticulate-rugose with transversal carina and two blunt tubercles.

**Wing:** Length of fore wing of 2.6 × its width, r:3-SR:SR1 = 23:23:87, vein SR1 of curved upward, vein CU1b exists, 1-CU1:2-CU1 = 16:55. Parastigma strong concave, cu-a not obvious and straight (Figure 2A).

**Feet:** Smooth and slender, hind femur robust, 3.8 × its maximum width. Hind tibia and tarsus 5.1 and 10.0 × its maximum width, respectively (Figure 2H).

**Metasoma:** Metasoma short oval, length 1.6 × its maximum width in dorsal view, and 2.5 × in lateral view (Figure 2D). Two oval-shaped maculae at metasoma basally, the ends curved inward. Length of metasoma cavity about 0.8 × metasoma (Figure 2E). The ovipositor is hidden under the metasoma.

**Color.** Black, antennae black, mandible brown except black basally, maxillary palpi and labial palpi brown, anterior, middle, and hind coxa and trochanter black, femur basally part black remaining brown, hind 5th tarsus black remaining brown. wing transparent and metallic, parastigma dark brown, veins of wing brown (Figure 1 and Figure 2).

**Male.** Similar to female, expect in having 28 antennomeres (Figure 1B).

**Distribution.** West and East Africa (Burkina Faso, Cameroon, Chad, DR Congo, Madagascar, Nigeria, Sudan, Togo, Kenya, and Tanzania), India.

Based on the results of morphological identification and the bar-code data from Agboyi et al., it is confirmed that it is the *Chelonus bifoveolatus* redescribed by De Saeger 1948. De Saeger and our description of the pigmentation on the hind foot is not consistent with Szepliget’s original description. Besides, our measurement is different from De Saeger’s who claimed that the length of tarsus is 7.0 × its maximum width.

### 3.2. Molecular Biological Identification

The sequencing of *Chelonus* sp. DNA barcode region of Cytochrome c Oxidase Subunit I (COI) was carried out, aiming at the identification of nucleotide sequences. With the improvement of the conditions of PCR, a 688 bp fragment of *Chelonus* sp. could be amplified repeatedly with the standard primer. The results of the search for similarities by BLAST tool in the Gen-Bank database was found that they all corresponded to the desired regions, which formed well-supported clusters in the NJ-tree. The results of the blast showed that the *Chelonus* sp. and *C. bifoveolatus* were obviously in the same branch, obtaining 99.99% similarity between them, besides, it also showed >99.69% similarity to *Chelonus formosanus* sonan 1932 (Figure 3).

The results of molecular biological identification, in combination with morphological evidence, indicated that *Chelonus* sp. was conspecific with *C. bifoveolatus.*

### 3.3. Performance on Host Eggs

#### Parasitism, Development Duration, and Longevity of *C. bifoveolatus* on FAW Eggs at Various Ages

From the initial egg masses, 109 (0-day-old), 121 (1-day-old), 141 (2-day-old) individuals hatched as the total number of larvae and 103, 116, and 129 of these successfully parasitized, respectively. No significant differences were found in the pupa duration, longevity of female and male for *C. bifoveolatus* on various age egg of FAW (pupa duration: *F*_2,12_ = 0.237, *p* = 0.792, longevity of female: *F*_2,12_ = 2.328, *p* = 0.140, longevity of male: *F*_2,12_ = 1.096, *p* = 0.365). Significant differences were found in the egg-larva duration on 0-, 1- and 2-day-old ages, the egg-larva duration on 2-day-old ages was significantly lower on 0-, 1-day-old (*F*_2,12_ = 9.418, *p* = 0.003). However, whatever age egg of FAW, the longevity of females was significantly longer on males (0-day-old: *t* = 8.996, *p* < 0.0001; 1-day-old: *t* = 11.841, *p* < 0.0001; 2-day-old: *t* = 7.037, *p* < 0.0001) (Table 1).

No significant differences were found in the parasitism, pupation, emergence, and female rate for *C. bifoveolatus* on various age egg of FAW (parasitism rate: *F*_2,12_ = 3.293, *p* = 0.072; pupation rate: *F*_2,12_ = 1.772, *p* = 0.212; emergence rate: *F*_2,12_ = 1.406, *p* = 0.283; female rate: *F*_2,12_ = 0.444, *p* = 0.652). The parasitism rate (95.75%), pupation rate (76.75%), and emergence rate (84.50%) were the highest on 1-day-old. The sex ratio of females to males was nearly 1:1 from all age egg of FAW (Figure 4).

## 4. Discussion

Based on morphological and genetic characteristics, the *Chelonus* spp. (egg-larval endoparasitoids) from parasitized eggs of *S. frugiperda* in Zambia were identified. Samples of *Chelonus* sp., were similar to *C. bifoveolatus* [50,51], which have previously been discovered from *Spodoptera* spp. in West and East Africa (Burkina Faso, Cameroon, Chad, DR Congo, Madagascar, Nigeria, Sudan, Togo, and Tanzania) [51,61,62], and was recently reported to be prevalent in Ghana and Benin [63]. However, it has been occasionally recovered from larvae of *Spodoptera exigua* (Hübner) in peri-urban vegetable onions gardens along the coast in Benin [64]. Interestingly, recovered *C. bifoveolatus* from Ghana and Benin had a >99% similarity to that reported in Zimbabwe and Kenya as well as from South Asia and Polynesia [63]. Our study indicated that although *Chelonus* sp. has a > 99% similarity with Ghana and Benin regions, it also has a >99% similarity to *C. formosanus* that is widespread distribution in Guangdong, Taiwan, Haina and Zhejiang (China) and some regions of India [65,66,67]. That means it is difficult to distinguish the two species by molecular biological using the universal primer COI, and it may be possible to use other primers to distinguish them [68]. However, the morphological comparison between the two *Chelonus* species provides strong evidence of differences in the size and shape of the spots at the basal of the mesosoma, as well as the transverse ridges of the propodeum (the results of our observations).

For most parasitoid species, the preference may be influenced by host age, host size and high-quality hosts [69,70,71,72]. For example, *Trichogramma mwanzai* and *Trichogrammatoidea lutea* parasitism significantly decreases with the increasing FAW host age [73]. Similar results were reported on *T. dendrolimi* that parasitized *Mythimna separata* (Walker) egg [74]. In our study, all stages of eggs of *S. frugiperda* (4–48 h after laying) are successfully parasitized by *C. bifoveolatus*, and the parasitism rate was >90%. In a previous study, *Chelonus inanitus* even could successfully parasitize all stages of eggs of *S. littoralis* (3 h old until 20 min before hatching), and parasitism rate ranged from 80% to 95% [75]. In contrast, *Chelonus.* sp. parasitism reaches 47% if old eggs of *Trichoplusia ni* Hubner (3 days old) were parasitized, where pseudoparasitization is frequent [76,77]. It may not be surprising that the latter parasitoid-host combination is induced in the laboratory, while the former is a naturally occurring parasitoid-host combination. The success of parasitism may also depend partly on where the parasitoid lays its eggs. The eggs of *T. dendrolimi* are laid in the host egg yolk [78]. On the contrary, *Trichogramma chilonis* are laid in or near the peripheral yolk of the host egg [79]. The eggs of *Trichogramma heliothidis* are generally oviposited next to the host embryo [80], while both are found within and outside the host embryo of *T. remus* [81], a parasitoid which is native to the Malay Peninsula, and due to the higher parasitism, has been used in augmentative biological control (ABC) programs against FAW in the Americas [82,83]. However, *C. inanitus* females lay eggs inside the embryo in mature hosts and inside the yolk in young hosts [84,85]. When parasitization occurs in young eggs, the parasitoid larvae must invest time and resources to reach the host embryo [86]. We also observed similar results when *C. bifoveolatus* parasitized all old eggs of *S. frugiperda*. In general, *Chelonus.* sp. is much larger than some egg parasitoids, thus the ovipositor may be inserted anywhere in the host egg, which may enable the parasitoid to successfully parasite eggs of all ages.

There are numerous associations between the genus *Chelonus* with many important *Spodoptera* species all over the world, such as *S. exigua, S. litura* and *S. eridania.* [76]. In our study, the egg-larva duration of *C. bifoveolatus* on 2-day-old ages host eggs was significantly lower on 0-, 1-day-old. Similarly, a previous study showed that the development time of *C. inanitus* reared on 75–80 h old eggs of the *S. littoralis* was significantly shorter than on 3–8 h old eggs [85]. This seems to be characteristic of all members of the genus *Chelonus* as similar observations were reported for *C. annulipes* [87], *C. curvimaculatus* [88], and for *C.* sp. [89]. In general, the genus *Chelonus* adapts to host development by varying its growth rate. Parasitoids that develop in non-feeding stages, such as eggs and pupae (represents a “closed” system), which can be assessed by the female at oviposition, regardless of any physiological changes that may occur later in the interaction [90,91]. In contrast, parasitoids exploit hosts that continue to feed and grow, such as larval stages (represents an “open” system), which vary with circumstances during the interaction [92,93]. The host adaptation mechanism of the egg-larval parasitoids may combine “closed” and “open” system strategies.

Life-history responses reflect the optimal balance between adaptation and constraint [94]. Host choice often reflects the koinobiont parasitoids female’s ability to find and oviposit in or on a host, rather than to provide optimum resources for immatures. Therefore, immature parasitoids may evolve ontogenetic responses to resource constraints.

## 5. Conclusions

In our study, *Chelonus bifoveolatus* accepted all test FAW eggs with different ages and completed development successfully. Duration of the egg-larva developmental stage of the *C. bifoveolatus* on 2-day-old ages host egg was significantly lower on 0- and 1-day-old. There were no significant differences in the parasitism rate, pupal rate, and emergence rate of *C. bifoveolatus* at various ages of FAW eggs. The longevity of female parasitoids was longer than male parasitoids, and the sex ratio of females to males was nearly 1:1. Generally, our results indicate that *C. bifoveolatus* performed well on various aged FAW eggs and is a potential biological control agent against FAW in Africa. However, the rearing of *C. bifoveolatus* on an alternative host should be considered for cost-effective mass production in the future.

## Figures and Tables

**Figure 1 insects-14-00061-f001:**
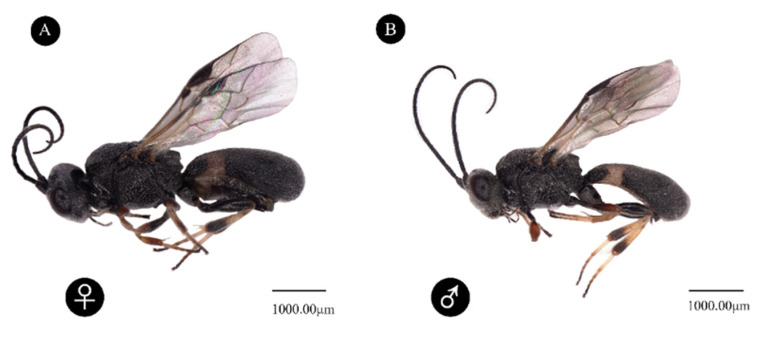
(**A**) *Chelonus* sp., female, holotype, habitus, lateral aspect, (**B**) male, holotype, habitus, lateral aspect.

**Figure 2 insects-14-00061-f002:**
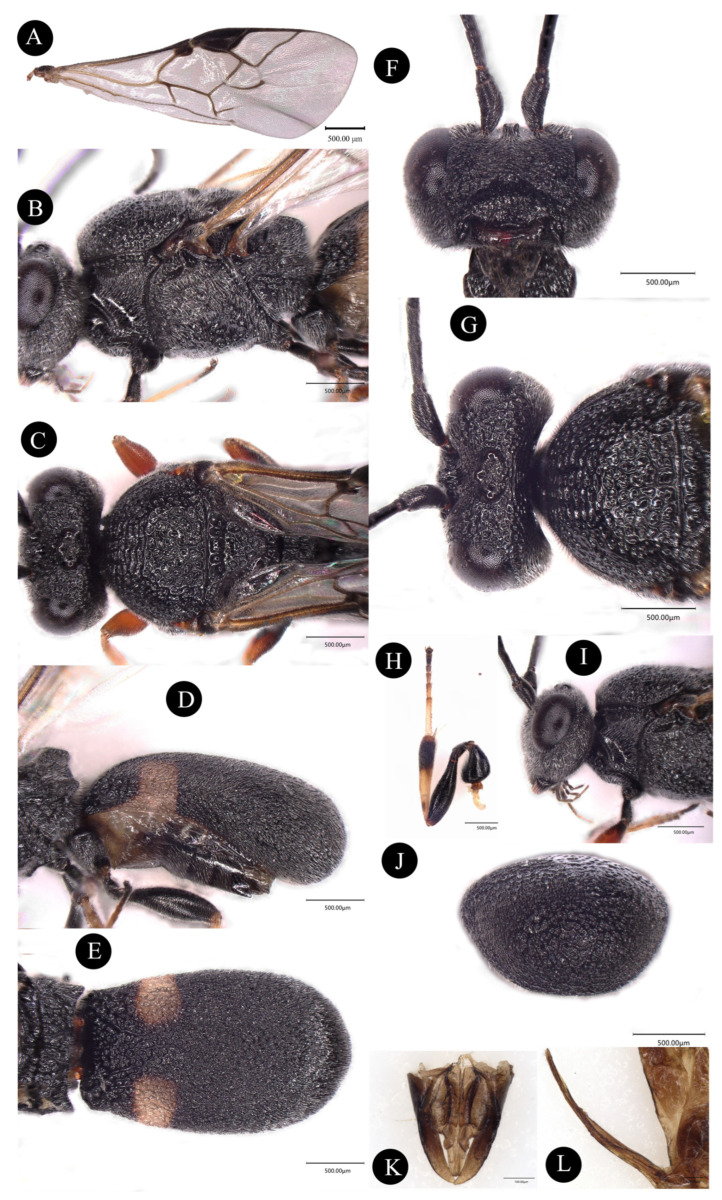
*Chelonus* sp., female, holotype. (**A**) fore wing, (**B**) mesosoma, lateral aspect, (**C**) mesoscutum, dorsal aspect, (**D**) mestasoma, lateral aspect, (**E**) metasoma, dorsal aspect, (**F**), head, anterior aspect, (**G**) head, dorsal aspect, (**H**) hind leg, lateral aspect, (**I**) head, lateral aspect, (**J**) metasoma, posterior aspect, (**K**) male genitalia, (**L**) female genitalia.

**Figure 3 insects-14-00061-f003:**
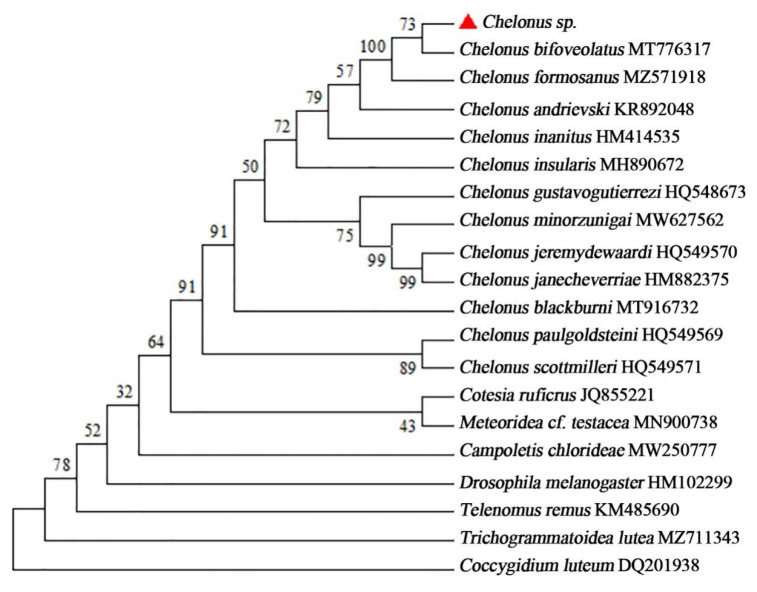
Molecular phylogenetic tree construction of *Chelonus* species based on the Neighbor-Joining method. The red triangle indicates the unidentified species.

**Figure 4 insects-14-00061-f004:**
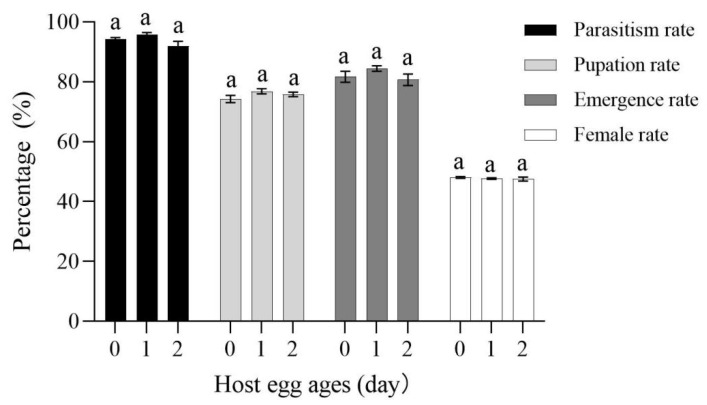
Data are means ± SE. Parasitism, pupation, emergence, and female rate of *C. bifoveolatus* parasite 24 h at various ages. (Different lowercase letters above a given group bars indicate significant difference (*p* < 0.05, ANOVA followed by Tukey’s test) on various ages.).

**Table 1 insects-14-00061-t001:** Comparisons of parasitism, developmental time and longevity of *C. bifoveolatus* on differently aged FAW eggs.

Host Eggs with Different Ages	Mean Egg Number per Egg Mass	Larvae Number Tested	Larvae Number Parasitized	Egg-Larva Duration (d)	Pupa Duration (d)	Longevity (d)	
						Female	Male
0 d	237.6	109.0 ± 8.9 a	102.8 ± 8.3 a	18.19 ± 0.18 a	12.10 ± 0.47 a	13.52 ± 0.26 aA	10.28 ± 0.25 aB
1 d	256.8	121.0 ± 6.9 a	115.8 ± 7.0 a	18.08 ± 0.19 a	12.05 ± 0.26 a	13.32 ± 0.28 aA	9.81 ± 0.09 aB
2 d	280.6	140.8 ± 9.8 a	129.0 ± 9.0 a	17.11 ± 0.21 b	11.77 ± 0.32 a	12.76 ± 0.24 aA	10.11 ± 0.29 aB

Note: Means ± SE are presented. Means in a column followed by the same lowercase letter do not differ significantly (*p* < 0.05) by Tukey’s HSD test and means in a row followed by the same uppercase letter do not differ significantly. (*p* < 0.05) by *T*-test.

## Data Availability

The data presented in this study are available on request from the corresponding author.

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
