# Peer review of "Identification of Chelonus sp. from Zambia and Its Performance on Different Aged Eggs of Spodoptera frugiperda"

_insects, 2023, doi:10.3390/insects14010061_

Round 1
Reviewer 1 Report
The work is conducted on a key egg-larval parasitoid associated to fall armyworm, a threat of global importance. The paper generally reads well except some language editing required throughout. I have also mentioned some critics on data analysis. Specific comments are provided below:
L15-16: please cross-check the consistency of the sentence
L22 and 23: should read Chelonus, please correct throughout
L44-47: sounds misleading info, please rephrase
L53-54: please use up to date info/literature, it should be far more than 28 countries affected
L69: should read “can pose”
L69-72: please what “damage” are you referring to; seems disconnected from previous statement; please rephrase or specify
L73: should read “alternative”
At this point, the manuscript requires a language revision; please correct all typos and carry out language revision on the text throughout.
L79-84: would be nice to group the species into egg, egg-larval, larval, pupal parasitoids
L112: again, please correct, should read, “the larvae were reared”
Section 2.1 please provide more narrative as how you rear the parasitoids
L138: remove one “centrifuged”
L162: remove the period after “respectively”
L164: I would use newly emerging parasitoids (e.g. less than 24 hours of age) to avoid testing those with significant previous oviposition experience
L165: delete “s” from eggs
L196-197: there is no mention whether the data in their raw form met the principle of normality and equal variance
L259: please italicize C. bifoveolatus
L292-294: please add citation
L311: the statement into brackets isn’t clear to me, please rephrase
L332-334: if you claim similarity between L332 and L333-334, therefore correct L332, the current statement looks confusing
L332-336: please enrich your discussion by articulating what could be the possible explanation
L352: please italicize “Chelonus”
L353-356: please cross-check the consistency of this
L484: should read “Hymenoptera”
L486: should read “taxonomic”
Please, correct typos on all references listed
Author Response
COMMENTS FOR AUTHORS
The work is conducted on a key egg-larval parasitoid associated to fall armyworm, a threat of global importance. The paper generally reads well except some language editing required throughout. I have also mentioned some critics on data analysis. Specific comments are provided below:
Answer: We thank the reviewer for the positive comments and suggestions; we have now included some clarifications in the revised manuscript to reflect the comments.
L15-16: please cross-check the consistency of the sentence
Answer: DONE. See line 15-16, page 1.
L22 and 23: should read Chelonus, please correct throughout
Answer: DONE. See line 22-23, page 1.
L44-47: sounds misleading info, please rephrase
Answer: DONE. Thank you very much for pointing out, we have redescribed this paragraph, see line 44-47, page 2.
L53-54: please use up to date info/literature, it should be far more than 28 countries affected
Answer: DONE. Thank you very much for sharing correct information about the number of Africa countries affected. We have redefined this paragraph, see line 52-53 page 2.
L69: should read “can pose”
Answer: DONE. See line 68, page 2.
L69-72: please what “damage” are you referring to; seems disconnected from previous statement; please rephrase or specify
Answer: DONE. The “damage” referring to excessive use of insecticides and small-holder farmers using scientifically unproven methods, we have rephrased this paragraph, see line 68-71, page 2.
L73: should read “alternative”
Answer: DONE. See line 72, page 2.
At this point, the manuscript requires a language revision; please correct all typos and carry out language revision on the text throughout.
Answer: DONE.
L79-84: would be nice to group the species into egg, egg-larval, larval, pupal parasitoids
Answer: DONE. We agree with your opinion, it is nice to group the species into egg, egg-larval, larval, pupal parasitoids, we have rephrased in the text, see line 77-85, page 2.
L112: again, please correct, should read, “the larvae were reared”
Answer: DONE. See line 113, page 3.
Section 2.1 please provide more narrative as how you rear the parasitoids
Answer: DONE. We provide more narrative as how to rear the parasitoids. See line 114-123, page 3.
L138: remove one “centrifuged”
Answer: DONE. See line 146, page 3.
L162: remove the period after “respectively”
Answer: DONE. See line 169, page 4.
L164: I would use newly emerging parasitoids (e.g. less than 24 hours of age) to avoid testing those with significant previous oviposition experience
Answer: Your suggestion is very pertinent and we agree with your suggestion, see line 171-172, page 4.
L165: delete “s” from eggs
Answer: DONE. See line 173, page 4.
L196-197: there is no mention whether the data in their raw form met the principle of normality and equal variance
Answer: DONE. Thank you very much for pointing this out, the number of parasitism is variables with different distribution than normal, but the percentage of emergence, percentage of pupation, developmental time and pro-portion of female offspring is variables with normal distribution, we have added sentences in the text, see line 204-213, page 5.
L259: please italicize C. bifoveolatus
Answer: DONE. See line 276, page 8.
L292-294: please add citation
Answer: DONE. See line 311, page 9.
L311: the statement into brackets isn’t clear to me, please rephrase
Answer: DONE. What we want to express the eggs of S. littoralis (3 h old until 20 min before hatching), we have rephrased in the text, see line 328-329, page 9.
L332-334: if you claim similarity between L332 and L333-334, therefore correct
Answer: DONE.
L332, the current statement looks confusing
Answer: DONE. We are sorry we didn't make that clear, we have rephrased in the text, see line 349-350, page 10.
L332-336: please enrich your discussion by articulating what could be the possible explanation
Answer: DONE. We agree with your opinion, we have added sentences to explain it in the text, See line 355-361, page 10.
L352: please italicize “Chelonus”
Answer: DONE. See line 367, page 10.
L353-356: please cross-check the consistency of this
Answer: DONE. See line 368-369, page 10.
L484: should read “Hymenoptera”
Answer: DONE. See line 504, page 13.
L486: should read “taxonomic”
Answer: DONE. See line 506, page 13.
Please, correct typos on all references listed
Answer: DONE.
Reviewer 2 Report
it is a good job, the remarks (minimal) are in the attached document.

Author Response
COMMENTS FOR AUTHORS
It is a good job, the remarks (minimal) are in the attached document.
Answer: We thank the reviewer for the positive comments and suggestions; we have finished the revision. Please see:
Page 1 – line 22-23, 25; Page 3 – line 102-103; Page 9 – line 308, 313;
Answer: DONE.
Page 3 – line 106. Support the procedure with a document that mentions Spodoptera breeding.
Answer: DONE. Thank you very much for pointing this out, we have provided the procedure with a document that mentions Spodoptera breeding. see line 123, page 3.
Page 2 – line 45-47.
Answer: We completely agree with your suggestion, and we have redescribed this paragraph. See line 44-47 page, 1-2.
Page 9 – line 293; Page 9 – line 306, 318; Page 10 – line 319, 346, 348;
Answer: Dear reviewer, we cross-checked and found that these species including Spodoptera exigua, Trichogramma mwanzai, Trichogramma chilonis, Trichogramma heliothidis, Chelonus blackburni and Chelonus ouclator, were first mentioned in the article so we did not abbreviate them.
Page 10 – line 351-352;
Answer: DONE. See line 367-368, page 10
Reviewer 3 Report
This paper presents the outcome of the morphological and molecular identification of a parasitoid of the fall armyworm collected in Africa (Zambia), as well as the results of a simple laboratory experiment assessing the effect of host egg age on parasitization success. The outcome of the identification is not surprising (the species has been repeatedly reported in Africa before) and the experiment shows that host egg age has no influence on parasitization. In short, there is not much novelty in this paper, and I leave it up to the editors to decide whether the content is worth publishing in Insects.
However, there are some more issues to be considered, which shed further doubt on the acceptability of the study and of the paper:
- The morphological identification lacks a diagnosis: Which literature sources were used to compare the different characteristics? Which characteristics of the studied material were in line with those given in previous descriptions of the species? Furthermore, there is no information whether and where type material was deposited, which is elementary for species descriptions.
- The laboratory experiment has very low replication: only 5 parasitoids were tested for each treatment (3 host egg ages); this means that the whole experiment comprised only 15 individuals - a very small experiment indeed.
- There are major problems with the writing of the paper and with the English language. There are plenty of grammatical and typing errors (too many to list) and several statements are confused or nonsensical (e.g. lines 41-44, 47-48, 92-93 in the Introduction). The paper would need a thorough rewrite to be acceptable for publication.
Some specific comments:
- line 124: there is no such thing as the "Braconid Guidebook". I assume the authors refer to the taxonomic literature on Braconidae
- lines 221-222: "rugose" is not a noun, but an adjective
- Table 1: it is not clear what is meant by "Parasitism number"; some of the column headings in this table refer to the host caterpillars, whereas others refer to the parasitoid. This should be improved.
- Figure 4: what are the error bars referring to?
- Parts of the Discussion are overly general. Paragraph 338-349 belongs to an Introduction section, rather than a Discussion
Author Response
Comments and Suggestions for Authors
This paper presents the outcome of the morphological and molecular identification of a parasitoid of the fall armyworm collected in Africa (Zambia), as well as the results of a simple laboratory experiment assessing the effect of host egg age on parasitization success. The outcome of the identification is not surprising (the species has been repeatedly reported in Africa before) and the experiment shows that host egg age has no influence on parasitization. In short, there is not much novelty in this paper, and I leave it up to the editors to decide whether the content is worth publishing in Insects.
Answer: We thank the reviewer for the pertinent comments and suggestions; Armyworm is one of the major agricultural pests in the world. The previous repeated reports of this parasitoid in Africa indicate that it is one of the most abundant parasitic natural enemies in the field. (Agboyi, L.K.; Goergen, G.; Beseh, P.; Mensah, S.A.; Clottey, V. A.; Glikpo, R.; Buddie, A.; Cafà, G.; Offord, L.; Day, R.; Rwomushana, I.; Kenis, M. Parasitoid complex of Fall Armyworm, Spodoptera frugiperda, in Ghana and Benin. Insects. 2020, 11, 68-83.) However, there is still a lack of relevant biological information so far. We thought it was worth the experiment, and it is not an end but a good beginning. Sometimes a story or a job may not turn out to be enough, but it always takes a person to contribute. As Tagore said, "there are no traces of birds in the sky, but I have flown over."
However, there are some more issues to be considered, which shed further doubt on the acceptability of the study and of the paper:
- The morphological identification lacks a diagnosis: Which literature sources were used to compare the different characteristics? Which characteristics of the studied material were in line with those given in previous descriptions of the species? Furthermore, there is no information whether and where type material was deposited, which is elementary for species descriptions.
Answer: DONE. Thank you very much for my advice. We re-supplemented those literature to support our results and what features of the material are consistent or inconsistent with previous descriptions of the species, in addition, adds some important information about the distribution of parasitoid, see line 253-259, page 7.
- The laboratory experiment has very low replication: only 5 parasitoids were tested for each treatment (3 host egg ages); this means that the whole experiment comprised only 15 individuals - a very small experiment indeed.
Answer: DONE. We very much agree with you. Although we only used 5 parasitic wasps to test 3 treatments, each replicate investigated more than 100 data (e.g. developmental time and longevity).
- There are major problems with the writing of the paper and with the English language. There are plenty of grammatical and typing errors (too many to list) and several statements are confused or nonsensical (e.g. lines 41-44, 47-48, 92-93 in the Introduction). The paper would need a thorough rewrite to be acceptable for publication.
Answer: DONE. We re-revised the grammatical and typing errors and several statements are confused or nonsensical in the article.
Some specific comments:
- line 124: there is no such thing as the "Braconid Guidebook". I assume the authors refer to the taxonomic literature on Braconidae
Answer: DONE. Thank you very much for pointing this out, we have re-revised the statements, see line 130-131, page 3.
- lines 221-222: "rugose" is not a noun, but an adjective
Answer: DONE. See line 233, page 6.
- Table 1: it is not clear what is meant by "Parasitism number"; some of the column headings in this table refer to the host caterpillars, whereas others refer to the parasitoid. This should be improved.
Answer: DONE. Thank you very much for pointing this out, we redefined “Parasitism number" as “Larvae number parasitized”, see Table 1.
- Figure 4: what are the error bars referring to?
Answer: DONE. Thank you very much for pointing this out, we have added sentences to explain it in the text, see line 300, page 9.
- Parts of the Discussion are overly general. Paragraph 338-349 belongs to an Introduction section, rather than a Discussion
Answer: DONE. Thank you very much for pointing this out, we have deleted in the text.